# Tryptophan-galactosylamine conjugates inhibit and disaggregate amyloid fibrils of Aβ42 and hIAPP peptides while reducing their toxicity

Ashim Paul [1], Moran Frenkel-Pinter [1], Daniela Escobar Alvarez[1], Giulia Milordini[2], Ehud Gazit [1], Elsa Zacco [2,3✉] & Daniel Segal[1,4✉]

Self-assembly of proteins into amyloid fibrils is a hallmark of various diseases, including Alzheimer's disease (AD) and Type-2 diabetes Mellitus (T2DM). Aggregation of specific peptides, like Aβ42 in AD and hIAPP in T2DM, causes cellular dysfunction resulting in the respective pathology. While these amyloidogenic proteins lack sequence homology, they all contain aromatic amino acids in their hydrophobic core that play a major role in their self-assembly. Targeting these aromatic residues by small molecules may be an attractive approach for inhibiting amyloid aggregation. Here, various biochemical and biophysical techniques revealed that a panel of tryptophan-galactosylamine conjugates significantly inhibit fibril formation of Aβ42 and hIAPP, and disassemble their pre-formed fibrils in a dose-dependent manner. They are also not toxic to mammalian cells and can reduce the cytotoxicity induced by Aβ42 and hIAPP aggregates. These tryptophan-galactosylamine conjugates can therefore serve as a scaffold for the development of therapeutics towards AD and T2DM.

[1] Department of Molecular Microbiology and Biotechnology, School of Molecular Cell Biology and Biotechnology, Tel Aviv University, Ramat Aviv, Tel Aviv 6997801, Israel. [2] The Maurice Wohl Clinical Neuroscience Institute, King's College London, Brixton, London SE5 9RT, UK. [3] RNA Central Lab, Center for Human Technologies, Istituto Italiano di Tecnologia, 16152 Genova, Italy. [4] Sagol Interdisciplinary School of Neuroscience, Tel Aviv University, Ramat Aviv, Tel Aviv 6997801, Israel. ✉email: elsa.zacco@iit.it; dsegal@post.tau.ac.il

Aberrant protein folding and consequent protein aggregation are hallmarks of a group of pathological conditions termed proteinopathies[1,2]. Despite arising from different genetic, environmental and regulatory factors, most proteinopathies exhibit sub-cellular and molecular similarities, characterized by the accumulation of misfolded proteins that often, self-assemble into thermodynamically highly stable amyloid fibrils, which are resistant to proteolytic degradation and lack functionally stable amyloid fibrils[3–7].

Alzheimer's disease (AD) and type 2 diabetes mellitus (T2DM) are prime examples of proteinopathies[2,6]. AD is the most common form of dementia in ageing populations[8,9]. A major triggering neuropathological cause of AD is the misfolding and accumulation of Aβ42, a hydrophobic peptide of 42-amino acids that is prone to self-assemble in the brain into toxic extra-cellular amyloid fibrils termed senile plaques[9,10]. Protein self-assembly is also a key feature in T2DM[11]. In this disease, pathological deposits of the peptide amylin, also called hIAPP, are found in the pancreatic islets of Langerhans of T2DM patients[11,12]. hIAPP is a 37-amino acid neuroendocrine peptide hormone produced by β-cells together with insulin[13,14]. When high insulin is required, such as in insulin deficiency, expression of hIAPP significantly increases, enhancing its propensity to aggregate into cytotoxic amyloid assemblies resulting in loss of β-cells and β-cell mass[14,15].

Epidemiological studies indicate comorbidity of AD and T2DM[16,17]. Hyperglycemia increases the chance of developing AD by at least 2-fold by contributing to the accumulation of amyloid plaques on brain lesions[16,18]. AD brains are less capable of glucose uptake from the surroundings, resembling a condition of insulin resistance[19]. In both T2DM and AD, oxidative stress is exacerbated, mitochondrial functions are impaired and neuron integrity compromised[20–23].

Intervening with the initial steps of protein misfolding and aggregation can thus be both a prophylactic measure and a means for modifying the course of these diseases[24,25]. In both Aβ42 and IAPP, the minimal aggregating sequence is a core of hydrophobic amino acids containing aromatic residues, KLVFFA and NFGAIL, respectively[26–28]. Aromatic amino acids have been identified as crucial in the formation of various amyloid structures. π-π stacking, rather than mere hydrophobicity, was shown to provide energy, order and directionality to promote amyloid assembly[29–31]. Based on these insights, peptide-based inhibitors were designed to contain aromatic residues whose side chains can intercalate the target amyloid aggregate and partially replace the original amino acids in π–π stacking[32–36]. Along the same lines, small aromatic molecules, e.g. polyphenols, have been amply studied as inhibitors of amyloid aggregation, where the aromatic ring appears to similarly interfere with amyloid self-assembly[37–41].

Combining the aromatic elements of amino acids and small molecules has also been attempted for amyloid inhibition. For example, a naphthoquinone–tryptophan hybrid was shown to be effective towards various amyloids (including Aβ42, IAPP, Tau, α-Syn) in vitro as well as in vivo[42–46]. However, a major limitation in the drugability of such molecules is their poor solubility in aqueous media[47–49]: Attaching a hydrophilic moiety for increasing solubility may enhance the overall therapeutic capacity of a drug[48]. Functionalization with glycans has been proven to enhance solubility, half-life and specificity of certain drugs in vivo[50]. The conjugation of glycans to peptides increases the overall interaction surface of the hybrid molecules, therefore increasing solubility by expanding the number of possible interactions with the solvent[50–53]. Along the same rationale, aromatic molecules, which display high potential as anti-amyloidogenic agents, but are by themselves hydrophobic, were conjugated to mannitol and glucosamine[43,54].

Here we examined this concept by synthesizing tryptophan-galactosylamine hybrid molecules and tested their ability to inhibit the aggregation of Aβ42 and hIAPP. Tryptophan was shown to be highly effective in intercalating the fibrils of various amyloidogenic proteins for inhibiting their aggregations[42,44,45,55,56]. Galactose (Gal), one of the most abundant monosaccharide in the human body, plays a vital role in numerous biological processes, modulating and mediating them[57,58]. To be able to speculate on the mechanistic role of galactose, we have included in the study GalNH$_2$ and GalNAc modifications at C2 of galactose influencing its charge density.

By means of Thioflavin T (ThT)-binding assay, circular dichroism (CD) spectroscopy, transmission electron microscopy (TEM) and Congo red birefringence, we show that the presence of galactose is key for the improved efficiency of tryptophan as amyloid inhibitor. The tested compounds are not toxic to mammalian cell lines. These results suggest that tryptophan-monosaccharide may be developed as disease-modifying therapeutics to target simultaneously Aβ42 and hIAPP.

## Results

**Inhibition of Aβ42 and hIAPP aggregation by hybrid molecules.** To examine the possibility of combining an amino acid and a glycan into a potentially improved amyloid inhibitor, we conjugated tryptophan to the N-glycoside 1-amino-1-deoxy-β-D-galactose (galactosylamine) and to its variants derived from the substitution of the OH on C2 with NH$_2$ and NAc groups. For simplicity, the hybrid molecules resulting from the conjugation of tryptophan and these sugars are hereafter referred to as WGal, WGalNH$_2$ and WGalNAc. As a negative control, the dipeptide formed by two tryptophan, WW was included. The structure of these hybrids is shown in Fig. 1.

**a** Aβ42 DAEFRHDSGYEVHHQKLVFFAEDVGSNKGAIIGLMVGGVVIA

**b** hIAPP KCNTATCATQRLANFLVHSSNNFGAILSSTNVGSNTY

**Fig. 1 Amyloidogenic peptides and aggregation inhibitors used in this study. a** Amino acid sequence of Aβ42. **b** Amino acid sequence of human hIAPP. **c** Structure of the dipeptide WW and of the tryptophan-galactosylamine hybrids WGal, WGalNH$_2$, and WGalNAc.

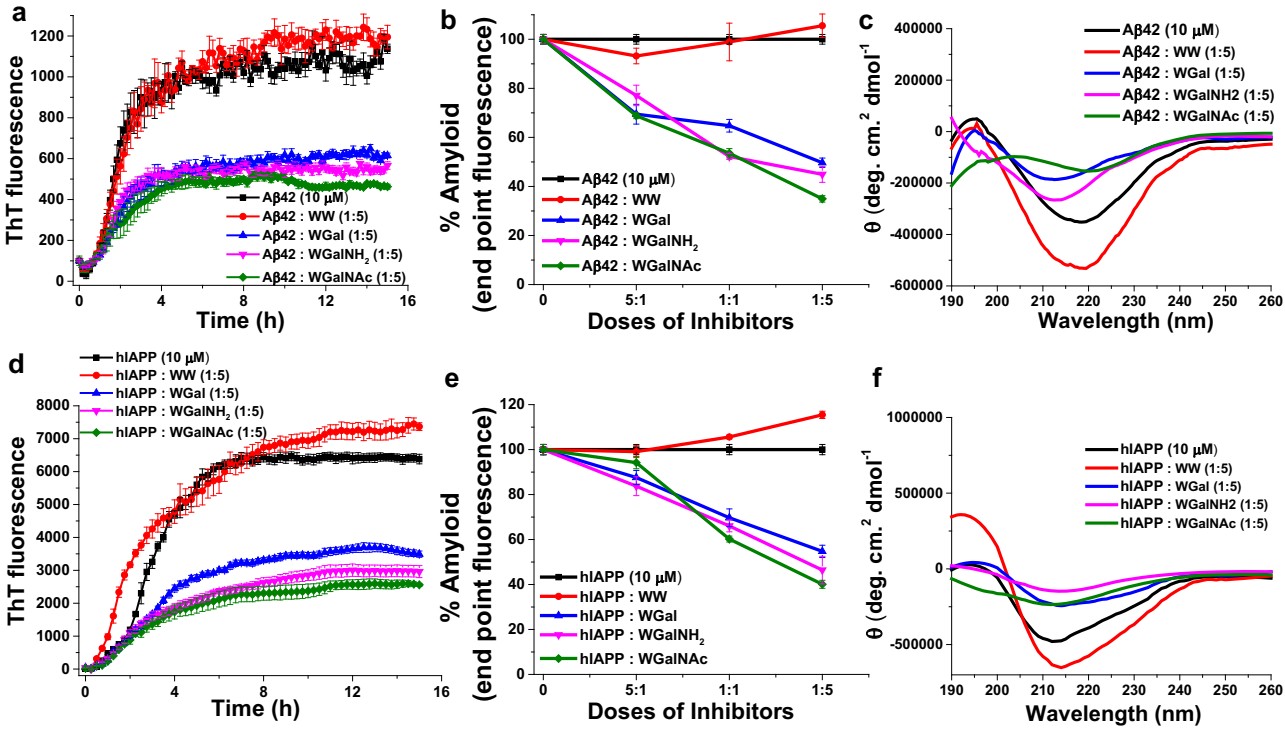

**Fig. 2 Inhibition of amyloid formation of Aβ42 and hIAPP by the tryptophan-galactosylamine hybrid molecules. a** ThT kinetics of Aβ42 aggregation in the absence or presence of 5-fold molar excess of the tested molecules. **b** Dose-dependent inhibition of Aβ42 aggregation by the tested molecules. **c** CD analysis of conformational changes of the Aβ42 aggregates in the absence or presence of the tested molecules. **d** ThT kinetics of hIAPP aggregation in the absence or presence of 5-fold molar excess of the tested molecules. **e** Dose-dependent inhibition of hIAPP aggregation by the tested molecules. **f** CD analysis of conformational changes of the hIAPP aggregates in the absence or presence of the tested molecules.

ThT was employed as an amyloid reporter dye to evaluate the effect of the tryptophan-galactosylamine hybrid molecules on amyloid formation by Aβ42 and hIAPP (Fig. 2). ThT is known to intercalate within the amyloid fibrils resulting in fluorescence emission proportionate to the amyloid growth. Under our experimental conditions (pH 7.4 and 37 °C), Aβ42 (10 μM) aggregation presented a sigmoidal curve, in agreement with previous studies[59,60]. The curve reached a plateau after ca. 8 h, indicating the completion of amyloid fibril formation. Next, Aβ42 was co-incubated with increasing doses of the tested hybrid molecules at peptide:hybrid molar ratios of 5:1, 1:1 and 1:5 (Fig. 2a, b and Supplementary Fig. 1). Considering the untreated Aβ42 plateau value as 100% aggregation, in the presence of 5-fold molar excess of WGal, WGalNH₂ and WGalNAc the level of amyloid fibrils was reduced by ~50%, ~57%, and ~70%, respectively (Fig. 2a, b). Inhibition of Aβ42 fibril formation by the hybrids was dose-dependent (Supplementary Fig. 1). WGal-NAc was the statistically most effective among the inhibitors (Supplementary Fig. 2a). In contrast, no inhibition effect was recorded for the dipeptide control WW and for isolated galactosylamine and tryptophan at any Aβ42:WW molar ratio (Fig. 2a, b and Supplementary Fig. 1a, e, f).

A similar inhibitory effect by the hybrid molecules was found toward hIAPP aggregation. In ThT kinetics, hIAPP was found to aggregate rapidly and reached plateau within ~6 h (Fig. 2d), indicating completion of amyloid fibrils formation in agreement with previous report[44]. Upon co-incubation of hIAPP with the hybrid molecules, a dose-response inhibition of hIAPP fibril formation was observed (Fig. 2d, e and Supplementary Fig. 3). Maximum inhibition of ~45%, ~52%, and ~64% by WGal, WGalNH₂ and WGalNAc, respectively, was recorded at 5-fold molar excess of the hybrid molecules (Fig. 2d, e and

Supplementary Fig. 3). Also, for IAPP, the marginally stronger inhibitory effect of WGalNAc was statistically significant (Supplementary Fig. 2b). No inhibitory effect was recorded in the presence of the controls WW, isolated galactosylamine and tryptophan (Fig. 2d, e and Supplementary Fig. 3a, e, f).

CD spectroscopy in the absence or presence of increasing doses of the hybrid molecules was carried out for Aβ42 and hIAPP to evaluate the effect of the conjugates on their conformation[44,61] (Fig. 2c, f and Supplementary Figs. 4, 5). Aggregated Aβ42 displayed a typical CD spectrum of β-sheet conformation with a single minimum at ~218 nm and a less intense maximum at around ~195 nm (Fig. 2c) as reported[62–66]. This indicated that upon aggregation the peptide transited from an initial disordered random coil structure to a highly organized β-rich conformation. The presence of the tested molecules during Aβ42 aggregation at a peptide:hybrid molar ratio of 5:1 did not alter the CD profile (Supplementary Fig. 4). However, the addition of equimolar amounts of WGal determined a diminished intensity minimum in the Aβ42 CD profile and a shifted band from 218 nm to 212 nm (Supplementary Fig. 4b). This effect was more pronounced at 5-fold molar excess of WGal, under which condition, the CD intensity was reduced markedly and the peak shifted from 218 nm to 211 nm (Fig. 2c), indicating a substantial reduction of β-sheet content as observed for other amyloid inhibitors of Aβ42[44,65,66]. A similar effect on the secondary structure of Aβ42 was observed in the presence of WGalNH₂ and WGalNAc (Fig. 2c and Supplementary Fig. 4c, d). No significant variation on the Aβ42 secondary structure was observed in the presence of the control WW (Fig. 2c and Supplementary Fig. 4a).

CD analysis was performed also for hIAPP (Fig. 2f and Supplementary Fig. 5). Aggregated hIAPP fibrils showed a positive peak at ~195 nm and a negative peak at ~213 nm,

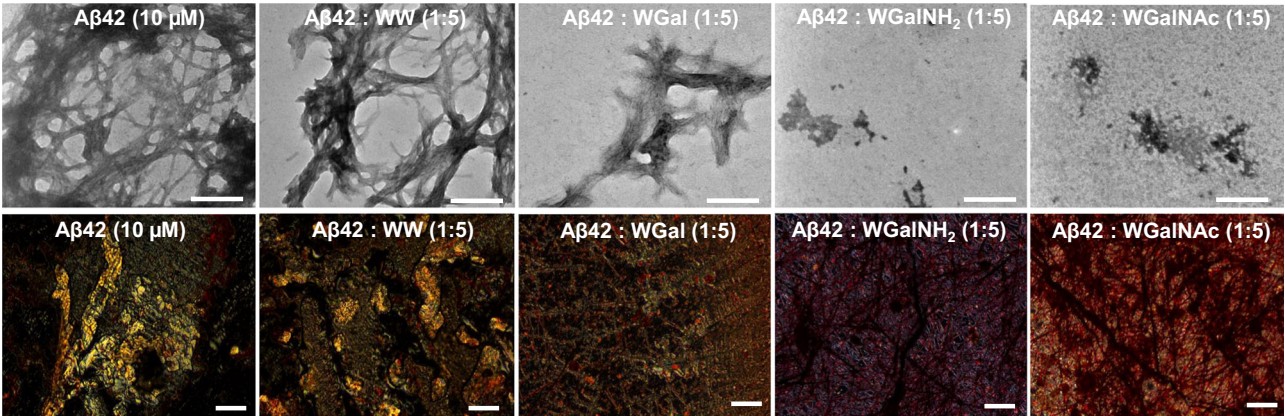

**Fig. 3 Analysis of Aβ42 fibrils in the absence or presence of the tryptophan-galactosylamine hybrid molecules.** TEM (upper panel) and Congo red stained birefringence (bottom panel) images of Aβ42 fibrils in the absence or presence of 5-fold molar excess of the tested molecules. Scale bars: 200 nm in TEM micrographs and 100 nm in Congo red birefringence images.

suggesting a β-sheet-rich conformation in agreement with previous reports[33,44,67,68]. With different degrees of effectiveness, the presence of all tested hybrid molecules caused a reduction of the intensity of the CD signal, indicating an overall lower of β-sheet content of hIAPP (Fig. 2f and Supplementary Fig. 5). WGalNH₂ and WGalNAc had a slightly more pronounced effect on the secondary structure of hIAPP compared to WGal (Fig. 2f and Supplementary Fig. 5). In the case of hIAPP, a substantial effect was also recorded by the control WW (Fig. 2f and Supplementary Fig. 5a): increasing doses of WW resulted in a proportionate increment of the intensity of the maximum at 195 nm and a slight increase of the intensity of the minimum, accompanied by a minor shift towards higher wavelengths, from 212 to 220 nm.

Next, we evaluated the effect of the hybrid molecules on the morphology and abundance of Aβ42 and hIAPP fibrils by means of TEM and Congo red birefringence[69–71]. In the absence of the hybrid molecules, Aβ42 formed long fibrils organized in characteristic packed bundles of different thickness, clearly visible in TEM micrographs (Fig. 3, upper panel)[44,72,73]. These fibrils also generated the amyloid specific gold-green birefringence upon staining with Congo red (Fig. 3, bottom panel and Supplementary Fig. 6)[74]. In the presence of 5-fold excess of WGalNH₂ or WGalNAc no fibrillar morphology was observed but small, amorphous aggregates appeared, indicating that these two hybrids significantly inhibited Aβ42 amyloid formation. In the presence of WGal, Aβ42 fibrillar morphology was still distinguishable, albeit the fibers were considerably fewer and smaller, indicating that this molecule was able to reduce the amount and the size of assemblies. In contrast, the control WW did not display any effect on the morphology of the Aβ42 amyloid fibrils at the same molar ratio (Fig. 3, upper panel). Similarly, no Congo red gold-green birefringence was observed when Aβ42 was incubated with WGalNH₂ or WGalNAc. In the presence of WGal we detected traces of birefringence, albeit minor, whereas intense gold-green birefringence was observed in the presence of the control molecule WW, indicating that it did not inhibit Aβ42 fibril formation (Fig. 3, bottom panel and Supplementary Fig. 6).

Similar analyses on hIAPP demonstrated that, in the absence of the hybrid molecules, hIAPP formed a thick network of amyloid fibrils, as shown by TEM (Fig. 4, upper panel), and exhibited golden birefringence upon staining with Congo red (Fig. 4, bottom panel and Supplementary Fig. 7) as reported[33,44,67]. When hIAPP was incubated with 5-fold molar excess of the hybrid molecules, no typical fibrillar assemblies were observed, rather some amorphous aggregates of different sizes were

detected (Fig. 4, upper panel). Likewise, no birefringence was observed, indicating inhibition of amyloid formation. In contrast, in the presence of the control WW, hIAPP amyloid fibrils were clearly visible in a more densely packed network and a clear gold-green birefringence was detectable. These results may suggest that not only WW was unable to inhibit the aggregation of hIAPP, but also that this dipeptide may promote further aggregation of hIAPP.

Collectively, the ThT, CD and TEM studies indicated that the tryptophan-galactosylamine hybrids were efficient in inhibiting aggregation and amyloid fibril formation of both Aβ42 and hIAPP. The control WW molecule lacked such inhibitory effect.

**Hybrid molecules disaggregate pre-formed peptides' fibrils.** For efficient treatment of proteinopathies it may be advantageous to disassemble amyloid aggregates/fibrils which are usually already present at the time of diagnosis[75–77]. Therefore, the ability of the hybrid molecules to disrupt pre-formed fibrils of Aβ42 and hIAPP was examined. To this end, Aβ42 and hIAPP peptides were allowed to aggregate for 10 h in the absence of the hybrid molecules. According to the ThT kinetic results, this time was sufficient to complete fibril formation (Fig. 2a, d). Then, various doses of the hybrid molecules were added to the pre-formed fibrils and the mixtures were incubated for additional 10 h. The process of disaggregation was monitored by ThT-binding assay, CD spectroscopy, TEM and Congo red staining (Fig. 5).

ThT fluorescence intensity remained at a plateau when Aβ42 was untreated, indicating that fibrils were maintained throughout the duration of the assay (Fig. 5a, b and Supplementary Fig. 8). Following addition of the hybrid molecules, the ThT signal dropped in a dose-dependent manner, reflecting a reduction in the amount of pre-formed fibrils (Fig. 5a, b and Supplementary Fig. 8). Maximum disruption, observed in the presence of 5-fold excess of the hybrid molecules, revealed ~40%, 52%, and 61% disaggregation in the presence of WGal, WGalNH₂, and WGalNAc, respectively (Fig. 5b and Supplementary Fig. 9a). In the presence of the control molecule WW, ThT fluorescence intensity was instead increased by ~20%, indicating that WW not only was unable to disaggregate the pre-formed fibrils, but also it may mediate the formation of new amyloid fibrils (Supplementary Fig. 9a). CD analysis (Fig. 5c and Supplementary Fig. 10) supported these findings. Untreated pre-formed Aβ42 fibrils exhibited a negative peak at ~214 nm and a positive maximum at ~194 nm, indicating β-sheet-rich conformation, whereas the intensity at ~214 nm was reduced in a dose-dependent manner in the presence of the hybrid molecules,

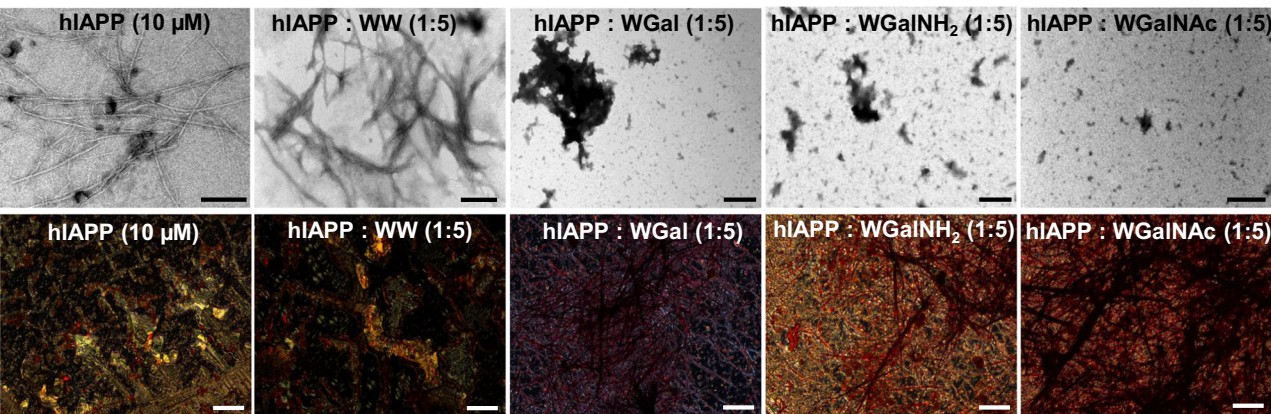

**Fig. 4 Analysis of hIAPP fibrils in the absence or presence of the tryptophan-galactosylamine hybrid molecules.** TEM (upper panel) and Congo red stained birefringence (bottom panel) images of hIAPP fibrils in the absence or presence of the 5-fold molar excess of the tested molecules. Scale bars: 200 nm in TEM micrographs and 100 nm in Congo red birefringence images.

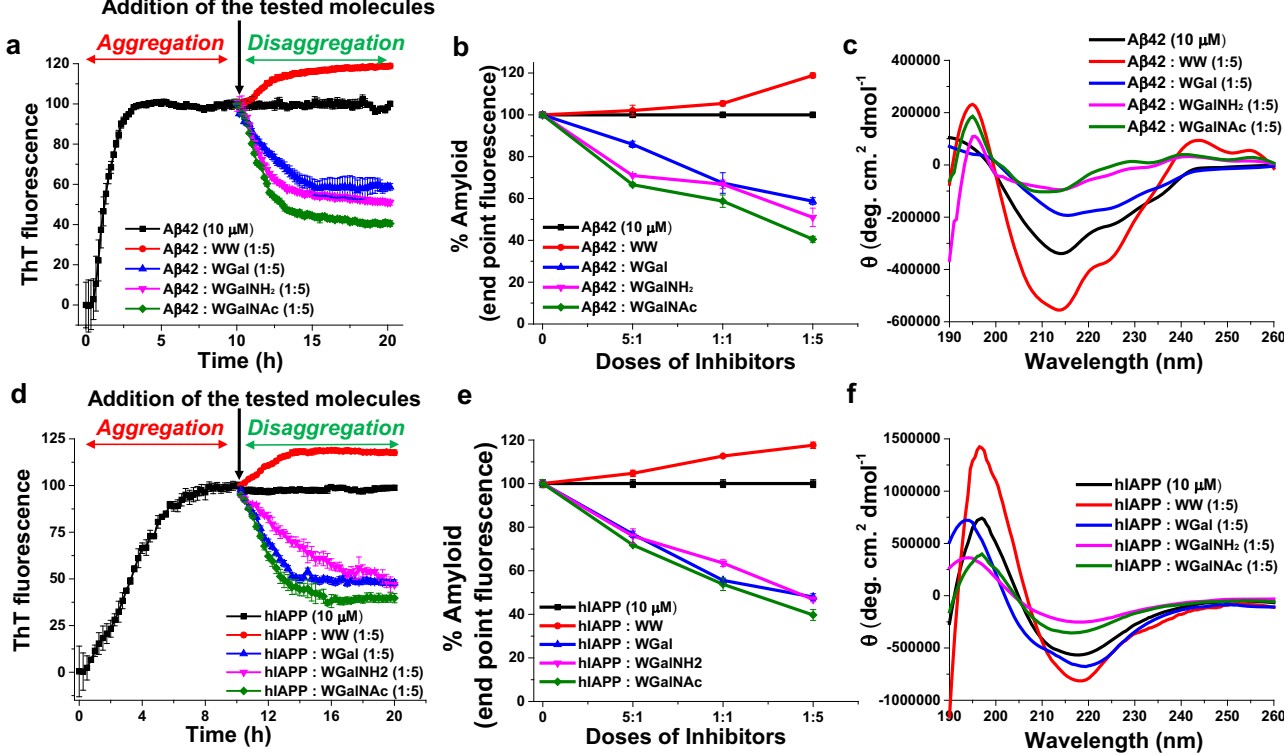

**Fig. 5 Effect of the tryptophan-galactosylamine hybrid molecules on pre-formed fibrils of Aβ42 and hIAPP. a** ThT kinetics for the disaggregation of pre-formed fibrils of Aβ42 in the absence or presence of 5-fold molar excess of the tested molecules. **b** Dose-dependent disaggregation of pre-formed fibrils of Aβ42 by the tested molecules. **c** CD analysis of pre-formed Aβ42 of in the absence or presence of the tested molecules. **d** ThT kinetics for the disaggregation of pre-formed fibrils of hIAPP in the absence or presence of 5-fold molar excess of the tested molecules. **e** Dose-dependent disaggregation of pre-formed fibrils of hIAPP by the tested molecules. **f** CD analysis of pre-formed hIAPP fibrils in the absence or presence of the tested molecules. Arrows in (**a**, **d**), indicate time of adding the tested hybrid molecules.

suggesting a substantial reduction in β-sheet content (Fig. 5c and Supplementary Fig. 10).

Comparable results were obtained for disaggregation of pre-formed hIAPP fibrils. (Fig. 5d, e and Supplementary Fig. 11). In the absence of the hybrid molecules, the ThT signal remains at a plateau, indicating that the presence of hIAPP fibrils in solutions during the assay (Fig. 5d, e and Supplementary Fig. 11). In contrast, in the presence of the hybrid molecules the ThT signal was reduced in a dose-dependent manner. Maximal reduction of signal, noticed at 5-fold excess of the hybrid molecules, represented ~53%, 53%, and 60% disaggregation in

the presence of WGal, WGalNH₂, and WGalNAc, respectively (Fig. 5e and Supplementary Fig. 9b). In contrast, a statistically significant 20% increase of ThT signal was observed in the presence of the control WW (Supplementary Fig. 9b). CD results corroborated these observations. Untreated pre-formed hIAPP fibrils exhibited a negative peak at ~217 nm and a maximum at ~196 nm, indicating β-sheet-rich conformation. In the presence of the hybrid molecules, the intensity at ~216 nm was reduced in a dose-dependent manner, suggesting a substantial reduction of β-sheet content (Fig. 5f and Supplementary Fig. 12).

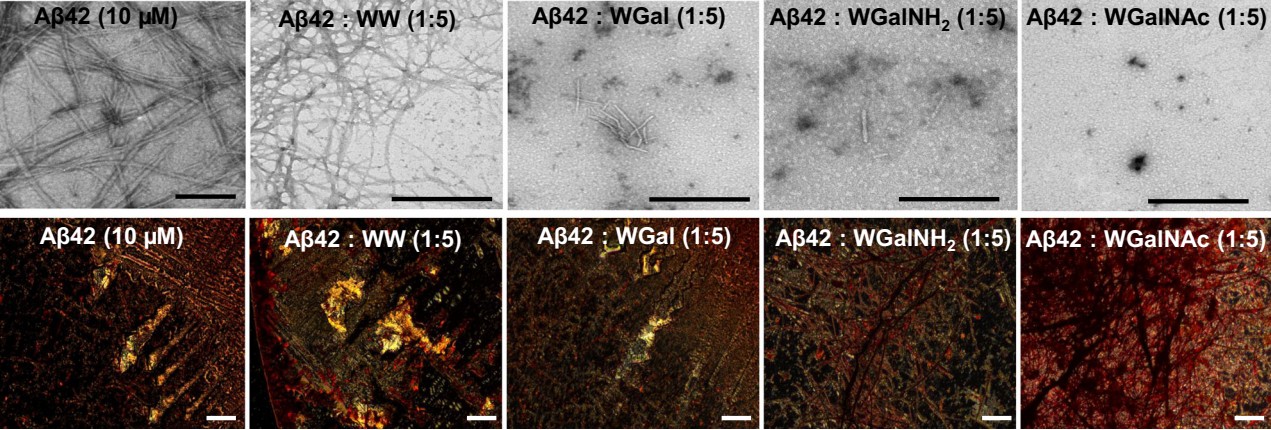

**Fig. 6 Analysis of pre-formed Aβ42 fibrils in the absence or presence of the tryptophan-galactosylamine hybrids.** TEM (upper panel) and Congo red stained birefringence (bottom panel) images of Aβ42 fibrils in the absence or presence of the 5-fold molar excess of the tested molecules. Scale bars: 200 nm in TEM micrographs and 100 nm in Congo red birefringence images.

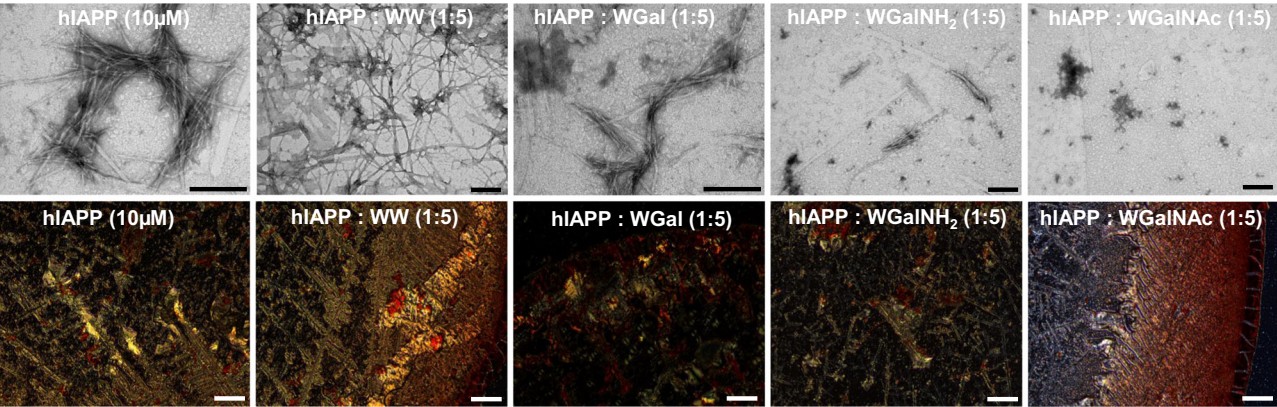

**Fig. 7 Analysis of pre-formed hIAPP fibrils in the absence or presence of the tryptophan-galactosylamine hybrid molecules.** TEM (upper panel) and Congo red stained birefringence (bottom panel) images of pre-formed hIAPP fibrils in the absence or presence of the 5-fold molar excess of the tested molecules. Scale bars: 200 nm in TEM micrographs and 100 nm in Congo red birefringence images.

TEM and Congo red assays lent further support to the ThT and CD results of the disaggregation experiments. Untreated pre-formed Aβ42 assemblies exhibited a fibrillar network indicative of the presence of amyloids (Fig. 6). However, in the presence of 5-fold excess of all tested hybrid molecules the fibril density was reduced, but the magnitude of this effect was different for each compound. No trace of fibrils was detected in the presence of WGalNAc, indicating an apparent transformation of the pre-formed fibrils into amorphous structures. WGalNH₂ was somewhat less efficient and WGal even less so (Fig. 6, upper panel). Congo red staining showed gold-green birefringence in untreated pre-formed Aβ42 fibrils (Fig. 6, bottom panel and Supplementary Fig. 13), yet no birefringence was observed when the pre-formed fibrils were incubated with WGalNH₂ or WGalNAc. However, in the presence of WGal or of the control WW faint birefringence was detected (Fig. 6, bottom panel and Supplementary Fig. 13).

Similar analyses of pre-formed hIAPP fibrils indicated that the untreated peptide exhibited dense fibrillar morphology and strong golden birefringence (Fig. 7); yet, upon treatment with WGal and WGalNH₂, the levels of amyloid fibrils and of Cong red birefringence were substantially reduced. No fibrils were detected by TEM in the presence of WGalNAc (Fig. 7, upper panel), and Cong red birefringence was markedly reduced, indicating complete disaggregation of pre-formed hIAPP fibrils (Fig. 7, bottom panel and Supplementary Fig. 14). In contrast, upon incubation with WW, fibril morphology and Congo red

birefringence were unchanged compared to the hIAPP peptide alone, suggesting that WW was unable to disaggregate the pre-formed fibrils of hIAPP (Fig. 7 and Supplementary Fig. 14).

Taken together, these results indicate an efficient disaggregation of the pre-formed assemblies of Aβ42 and hIAPP by the three hybrid molecules, among which WGalNAc was found to be the most effective.

**Hybrid molecules reduce Aβ42 and hIAPP-induced toxicity.** The effect of the tryptophan-galactosylamine hybrid molecules toward cytotoxicity caused by Aβ42 and hIAPP amyloids was evaluated. To this end, we first tested whether the hybrid molecules carried any cytotoxicity themselves. We incubated them at various concentrations (1–250 μM) with either human neuroblastoma (SH-SY5Y) or with human embryonic kidney (HEK-293) cell lines and assessed cell viability by XTT assay[33,44]. No considerable toxicity (≥95% viability) was caused by the hybrid molecules towards either cell line, even at the highest concentration of 250 μM (Fig. 8). However, the control molecule WW exhibited a slightly toxic effect at this concentration, and ~88% and ~83% viability was observed for SH-SY5Y and HEK-393 cells, respectively.

To examine the effect of the hybrid compounds on the cytotoxicity caused by Aβ42 and hIAPP aggregates towards these cell lines, Aβ42 and hIAPP peptides were first incubated, separately, at various concentrations (1–20 μM) to allow fibril

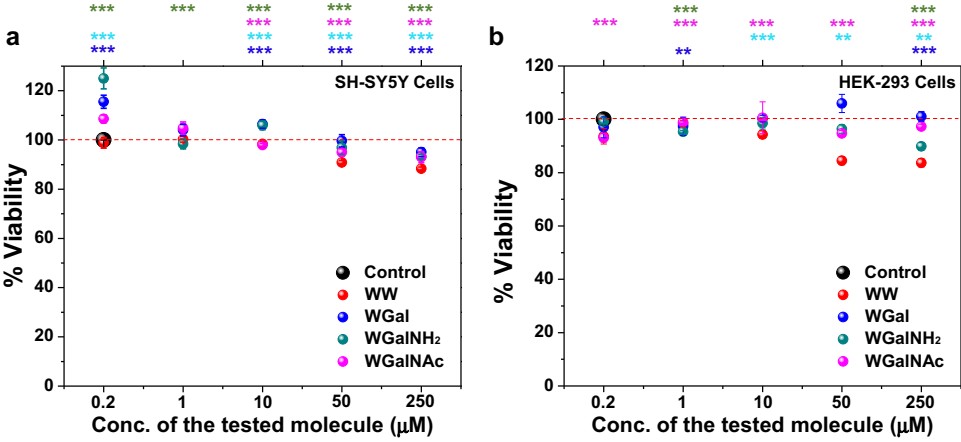

**Fig. 8 Evaluation of cytotoxicity of the tryptophan-galactosylamine hybrid molecules.** The toxicity of the hybrid molecules towards (**a**) SH-SY5Y and (**b**) HEK-293 cells evaluated by XTT assay. Cells were treated with various concentrations (1–250 μM) of the hybrid molecules and incubated for 24 h. **p < 0.005 and ***p < 0.001 compared to untreated samples. Untreated cells were considered as 100% viable.

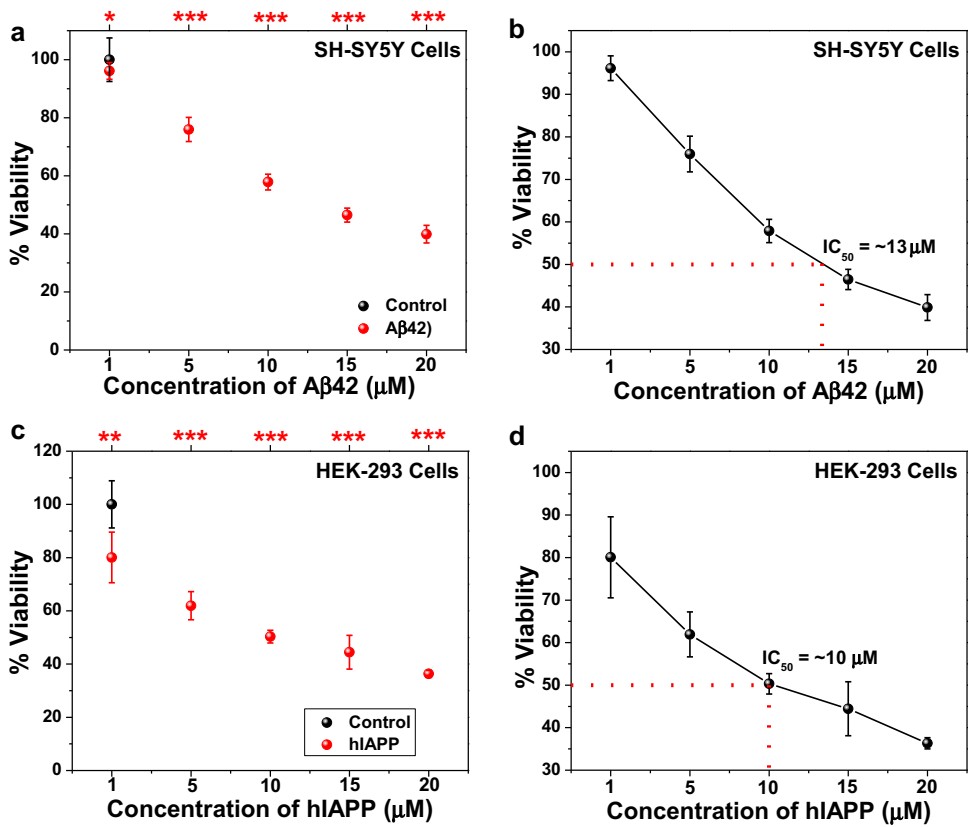

**Fig. 9 Evaluation of IC$_{50}$ for Aβ42 and hIAPP aggregates. a** Effects of pre-formed Aβ42 fibrils on viability of SH-SY5Y cells were examined using XTT reduction assay. **b** IC$_{50}$ value of Aβ42 assemblies was calculated from the XTT assay and found as ~13 μM. **c** Effects of pre-formed hIAPP fibrils on viability of HEK-293 cells were examined using XTT reduction assay. **d** IC$_{50}$ value of pre-formed hIAPP fibrils was calculated from the XTT assay and found as ~10 μM. Confluent cells were incubated with various concentrations (1–20 μM) of pre-formed Aβ42 and hIAPP fibrils (generated based on ThT results). After 24 h incubation with the peptides, cell viability was determined by XTT reduction assay. **p < 0.05, **p < 0.005 and ***p < 0.001 compared to untreated cells. Untreated cells were considered as 100% viable.

formation. The peptide assemblies were then incubated for further 24 h with SH-SY5Y or HEK-293 cells and cell viability was monitored by XTT assay (Fig. 9). The calculated IC$_{50}$ values for Aβ42 and hIAPP were found to be ~13 μM and ~10 μM, respectively, as compared to untreated control cells, in agreement with previous reports[44,78,79]. Various doses of

the hybrid molecules (Aβ42/hIAPP:hybrids = 50:1, 20:1, 10:1, 5:1, 1:1, 1:5, 1:10, 1:20) were added separately to pre-aggregated Aβ42 (13 μM) or hIAPP (10 μM). The mixtures were then independently applied to the respective cells for 24 h, and cell viability was further measured by XTT assay (Fig. 10a).

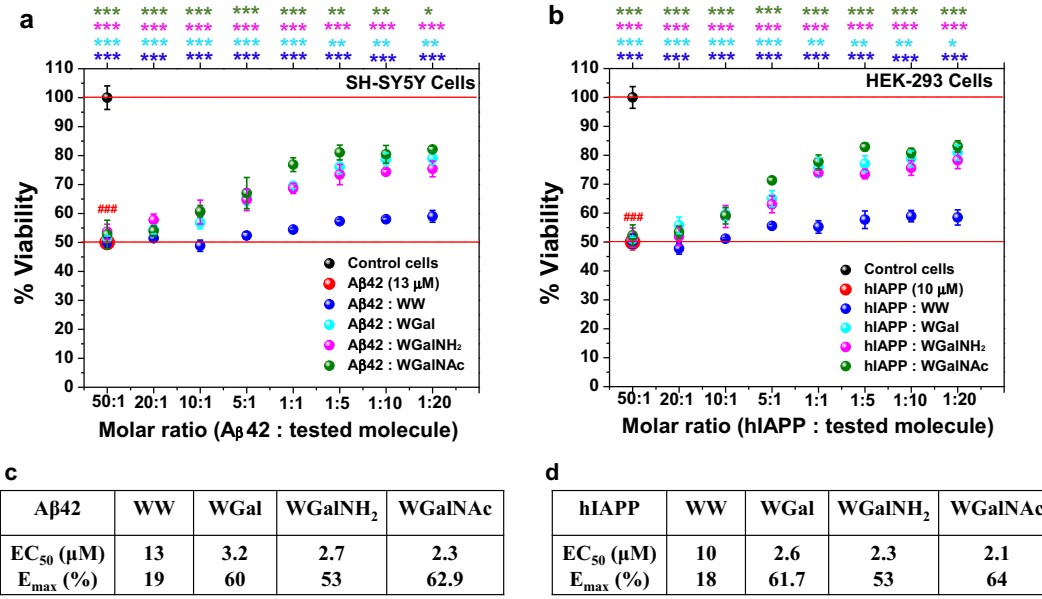

**Fig. 10 Effects of the tryptophan-galactosylamine hybrid molecules towards cytotoxicity of pre-formed amyloids.** (**a**) Aβ42- and (**b**) hIAPP-induced cytotoxicity in the presence or absence of various doses of the tested molecules were evaluated using XTT reduction assay. $EC_{50}$ and $E_{max}$ of the tested molecules when applied to reduce the cytotoxicity induced by pre-formed fibrils of (**c**) Aβ42 and (**d**) hIAPP. $^{###}p < 0.001$, compared to the untreated cells. $^{**}p < 0.05$, $^{**}p < 0.005$ and $^{***}p < 0.001$ compared to Aβ42 (**a**) and hIAPP (**b**) treatment. Untreated cells were considered as 100% viable.

All of the tested hybrid molecules, including the control WW, significantly reduced the toxicity of pre-formed Aβ42 fibrils in a dose-dependent manner. For example, 20-fold excess of WW, WGal, WGalNH$_2$ and WGalNAc caused an increment in viability of cells, incubated with pre-aggregated Aβ42, of 19%, 60 ± 4%, 53 ± 5%, and 62.9 ± 2%, respectively (maximum effect, or $E_{max}$) (Fig. 10a and Supplementary Fig. 15). In agreement with the in vitro assays, in the cell-based assays WGalNAc gave the highest effect, with an $EC_{50}$ of 2.3 μM. WW, WGal and WGalNH$_2$ resulted in $EC_{50}$ of 13, 3.2 and 2.7 μM, respectively (Fig. 10a, c and Supplementary Fig. 15a).

Similarly, when HEK-293 cells were exposed to pre-formed hAIPP fibrils, in the absence or presence of various doses of the hybrid molecules, cell viability was found to be increased proportionately to the dose of the hybrid molecules. 20-fold excess of WW, WGal, WGalNH$_2$ and WGalNAc incremented the viability of 18 ± 5%, 61.7 ± 5%, 53 ± 5%, and 64 ± 4% respectively (maximum effect, or $E_{max}$) (Fig. 10b, d and Supplementary Fig. 15b). The calculated $EC_{50}$ was 10, 2.6, 2.3, and 2.1 μM, respectively (Fig. 10b, d and Supplementary Fig. 15b).

These results indicated that the hybrid molecules significantly reduced the cytotoxic effect induced by Aβ42 and hIAPP fibrils. WGalNAc was the most effective among them.

## Discussion
In this work, we evaluated the potential of a new class of highly-soluble tryptophan-galactosylamine hybrid molecules to alter the progression of the aberrant aggregation of the disease-associated peptides Aβ42 and hIAPP, both in vitro and in mammalian cell lines.

The amino acid tryptophan was chosen in this study because it was reported to display the most amyloidogenic propensity among the proteinogenic amino acids, suggesting a potentially strong involvement of its aromatic side chain in the molecular associations leading to the formation of amyloid fibrils[55,80]. Indeed, tryptophan has been shown to be directly involved in the abrogation of protein aggregation by targeting the aromatic

recognition interfaces of amyloidogenic proteins[42]. Tryptophan derivatives have been employed in the design of several inhibitors of amyloid formation. For example, tryptophan-containing peptides have been employed to reduce Aβ42-derived toxicity in animal models[81]. Tryptophan-coated gold and silver nanoparticles have been shown to inhibit both spontaneous and seed-induced aggregation of insulin[82]. Naphthoquinone–tryptophan hybrids displayed promising potential for mitigating the amyloidogenicity of several proteins and peptides[44,45,83].

The present work aimed at improving the drugability of tryptophan by combining its ability to target amyloid species with a component that would confer the capacity to dissolve the target protein aggregates. Thus, we postulated that the conjugation of tryptophan to a glycan unit could increase solubility and target selectivity of the hybrid molecule[50,84,85].

The glycan galactose is involved in numerous biological processes, hence galactosylation is considered as a useful tool for delivery of therapeutics[86–88]. For example, galactosylated nano-carriers have been shown to improve site-specific delivery of siRNA and anticancer drugs[86]. Indeed, galactose has been employed in studies for the treatment and diagnosis of several diseases, as well as vaccine development[89–91]. D-galactose, together with D-glucose, are actively transported across the blood-brain barrier (BBB), making these hexoses highly attractive for the specific delivery of drugs to the brain[92]. Galactose-conjugated dopamine was found to have better BBB permeability, bioavailability, and therapeutic effects against Parkinson's disease[93–95]. Finally, galactose is a preferred glycan candidate for targeting hIAPP because, due to hepatic metabolism, insulin and glucose level responses to galactose are lower than for glucose[96,97].

Taken together, this evidence points to tryptophan and galactose as desirable building blocks for the design of potential hybrid molecules against amyloid formation.

At the core of the amyloidogenic process there are conformational changes of the native, soluble peptides into unstable oligomers that subsequently assemble into insoluble protofibrils and fibrils[98]. The formation of oligomers in the absence of other pre-existent larger peptide species is refer to as primary

nucleation[99]. These oligomers can grow into fibers by the addition of other oligomers (elongation) or monomers (secondary nucleation)[60]. The kinetics of amyloid formation generally follows a sigmoidal trend, in which a stable lag phase, an exponential phase and a plateau, the stage by which no more monomers are available, can be distinguished[100,101]. According to the mechanism of action of a potential inhibitor, any of the phases of the aggregation process may be effected. All our in vitro analyses demonstrated an appreciable amyloid inhibitory effect on both Aβ42 and hIAPP by all the tryptophan-galactosylamine hybrids. The data collected for WGal, WGalNH$_2$ and GalNAc show that there is no kinetic effect caused by the inhibitors on primary nucleation or elongation (lag phase or exponential phase). Instead, the plateau levels of both Aβ42 and hIAPP are significantly reduced by the presence of the hybrid molecules. To speculate on the mechanistic role that these inhibitors are playing on the aggregation of the target peptides, it is important to bear in mind that there are many ways by which small molecules may model the aggregation pathway[102,103]. Among these, it has been reported that various inhibitors reduce the amount of amyloids by redirecting the aggregation towards alternative off-path products, rather than preventing the self-assembly of the monomers[41,104–106]. We believe that this is the case for the hybrid molecules studies here: the lack of an obvious kinetic effect on primary/secondary nucleation or elongation suggests the sequestration of the monomers and oligomers to a parallel alterative pathway that does not lead to amyloid formation but to smaller and often amorphous assemblies. This hypothesis is supported by the ca. 50% reduction of ThT-positive aggregates in the presence of the inhibitors; by the variation in Aβ42 and hIAPP's secondary structures; by the lack of fiber-like assemblies as shown by TEM and Congo red birefringence; and by the significant lower cellular toxicity.

A closer look into the results of the ThT-binding assays suggests that although the differences between the effects of the inhibitors are small, WGalNAc is on average a significantly stronger amyloid inhibitor compared to WGal and WGalNH$_2$, as indicated by the $p$-value calculations (Supplementary Figs. 2, 9). The superiority of WGalNAc may be accounted for by its acetyl group. This moiety may actively contribute to increase the energetically favorable CH-π stacking between the cluster of carbon atoms of the hexose ring and the target peptides[107,108]. Moreover, the larger molecular radius of –NAc compared to –NH$_2$ and –OH may allow WGalNAc to interact more stably with a higher number of atoms of the target peptides[109]. Finally, the slightly more hydrophobic acetyl group may also participate in binding the amyloidogenic peptides tested, and lead the whole hybrid molecule WGalNAc towards a more balanced equilibrium between the ideal hydrophilicity and hydrophobicity[110].

Alongside the designed hybrid molecules, all experiments performed in this work included the dipeptide WW, here referred to as "negative control" which allowed to assert the effect of the galactosylamine derivatives in comparison to the tryptophan alone. However, in some circumstances, WW displayed a stimulatory effect on the aggregation of Aβ42 and hIAPP. This behavior became more pronounced and statistically significant (Supplementary Figs. 2, 9) during the disaggregation of pre-formed fibrils (Figs. 5–7). These results could be explained by the amyloidogenic nature of tryptophan and by its ability to self-assemble into fibrils, resulting in an additive aggregation level[55,80]. The co-presence of aggregating peptides may have functioned as further nucleation agent for the π–π stacking of tryptophan, which, on its end, could also have consequently induced further Aβ42 and hIAPP aggregation into mature amyloid fibrils. This effect, however, did not translate into an increment of toxicity in the in-cell assays: albeit not as efficiently, WW, like the hybrid molecules, reduced the cytotoxicity induced by extra-cellular pre-aggregated Aβ42 and hIAPP. To explain the partial inhibitory effect of WW on Aβ42 and hIAPP-induced toxicity, we postulate that WW may have driven the aggregates of these peptides towards larger bundle of fibrils, which are reported to be less toxic than smaller oligomeric forms[111–114]. WW's effect on cytotoxicity, however statistically significant, is very weak when compared to the hybrid molecules: EC$_{50}$ of all hybrids are between 2.5 and 5-fold lower than the one calculated for WW.

Significant breakthroughs in inhibition of diseased associated amyloid aggregation have led to different therapeutic strategies, which include targeting the protein of interest with either small molecules or antibodies[115–117]. Despite the efforts and the sophisticated specificity offered by anti-aggregation antibodies, these have so far failed at clinical trials because of the aberrant immune response they initiate[118,119]. In this sense, small molecules offer substantial advantages over antibodies, since they are immunologically tolerated, more stable in the cellular environment, and often offer higher potential of crossing the BBB[120].

In this work we presented anti-aggregating tryptophan-galactosylamine hybrid small molecules. We show a dual effect of these small molecules, capable of both reducing and reversing the formation of amyloid fibrils. Moreover, these molecules are non-toxic toward mammalian neuroblastoma and kidney cell lines and are able to significantly reduce the cytotoxicity induced by pre-formed fibrils of Aβ42 or hIAPP. Collectively, the results here reported indicate that the hybrid system could be employed as a scaffold in the development of therapeutics towards multiple proteinopathies. Further studies will follow to obtain deeper insight on the mechanism of action of these hybrid molecules and to chemically improve them for higher efficacy and in vivo validation. Amyloids of Aβ42 and hIAPP co-exist in major age-related diseases. Thus, the hybrid molecules could potentially offer single disease-treating therapeutics to these disorders.

## Methods

**Materials and sources**. Aβ42 was purchased from rPeptide (USA). hIAPP was purchased from Peptide 2.0 (USA). The inhibitors (WGal, WGalNH2 and WGalNAc) were purchased from Aldlab (USA). WW was purchased from GL Biochem Shanghai (China). Lipids were purchased from Avanti Polar Lipids (USA). All chemicals and reagents were of analytical grade. Unless otherwise stated, all chemicals were obtained from Sigma-Aldrich (Rehovot, Israel).

**Stock preparation**. Aβ42 and hIAPP were monomerized by a 10 min pre-treatment with HFIP and the solvent was evaporated using a Speed Vac. The resulting thin films were dissolved in PBS (100 mM, pH 7.4) and sonicated for 5 min to get a stock concentration of 50 μM, which immediately diluted to required concentration for an experiment. Stock solutions of ThT (4 mM in PBS) and hybrid molecules (10 mM in PBS) were prepared. All the stock solutions were diluted according to the requirement.

**Thioflavin T-binding assay**. For the aggregation kinetics, the stock solutions of Aβ42 or hIAPP were diluted in a 96-well black plate so that the final mixture (100 μl) contained 10 μM peptide and 20 μM ThT in 100 mM PBS. The experiment was performed according to the previous report[44]. For fibril disassembly assays, Aβ42 and hIAPP were first allowed to self-assemble until a ThT-associated plateau was reached; the hybrid molecules were then added separately to designated wells (peptide: hybrid molecule = 5:1, 1:1, 1:5), and the fluorescence values were further followed until a new plateau was reached. Kinetics fluorescence data were collected at 37 °C in triplicate using Infinite M200 microplate reader (Tecan, Switzerland), with measurements acquired at 15 min intervals. Excitation and emission wavelengths were set at 440 nm and 485 nm, respectively.

**Circular dichroism spectroscopy**. To analyze the secondary structure of the aggregated or disassembled peptides, 300 μL samples (10 μM of Aβ$_{42}$ and hIAPP) were placed in a quarz cuvette (path length 1 mm) and CD spectra were recorded on a Chirascan spectrometer between 190–260 nm. The blank was subtracted from the acquired CD spectra. For the CD study in the presence of the tryptophan-galacosylamine molecules, the spectra of hybrid molecules were also recorded separately and subtracted from the respective sample spectra. The normalized data were plotted using Origin pro 2015 software.

**Transmission electron microscopy**. Samples (10 μL) were placed for 2 min on 400-mesh copper grids covered with carbon-stabilized Formvar film (Electron Microscopy Sciences (EMS), Hatfield, PA). Excess fluid was removed, and grids were negatively stained with 2% uranyl acetate solution (10 μL) for 2 min. After excess fluid removal, the samples were visualized using a JEM-1400 TEM (JEOL), operated at 80 kV.

**Cell cytotoxicity experiments**. The SH-SY5Y and HEK-293 cell lines ($2 \times 10^5$ cells/mL) were cultured in 96-well tissue microplates (100 μL/well) and allowed to adhere overnight at 37 °C. The conjugate molecules were dissolved in DMEM: nutrient mixture F12 (Ham's) (1:1) (Biological Industries, Israel) at different concentrations. The negative control was prepared as medium without hybrid molecules and treated in the same manner. 100 μL of medium with or without hybrid molecules were added to each well in triplicate. Following incubation for 24 h at 37 °C, cell viability was evaluated using the 2,3-bis(2-methoxy-4-nitro-5-sulfophenyl)-2H-tetrazolium-5-carboxanilide (XTT) cell proliferation assay kit (Biological Industries, Israel) according to the manufacturer's instructions. Briefly, 100 μL of activation reagent was added to 5 mL of XTT reagent, followed by the addition of 50 μL of activated-XTT solution to each well. After 2 h incubation at 37 °C, color intensity was measured using an ELISA microplate reader at 450 nm and 630 nm. Results are presented as mean and mean standard error. Each experiment was repeated minimum three times.

**Statistics and reproducibility**. ThT-binding assays and CD spectrophotometric assays presented in this work were performed at least five times, each time with a minimum of three replicates per condition. Between 8 and 10 representative TEM micrographs and Congo red stained images were acquired at the end of each ThT-binding assay, both for aggregation inhibition and aggregate disruption experiments. XTT viability assays were repeated five times, each time with triplicates for each molecule and concentration.

For the quantitative data derived from ThT and XTT assays, statistical significance, expressed as p-value, was determined using Student's t-test calculation applied at the end of the experiments, setting tails to 2 (tailed distribution) and type to 2 (samples equal variance), according to reported methods[121]. From the XTT viability assays, $EC_{50}$ and $E_{max}$ values were determined from the dose-response curves of each tested molecule using non-linear least-square data fitting, according to literature[122].

**Reporting summary**. Further information on research design is available in the Nature Research Reporting Summary linked to this article.

## Data availability
The data associated with the figures presented on this article are available as excel files in the Supplementary Data. Further details generated and analyzed during the current study are available from the corresponding authors upon reasonable request.

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

## Acknowledgements
This research was partially supported by the Alliance Family Trust (to D.S). E.Z. wishes to acknowledge the Daniel Turnberg Travel Fellowship, 2017–2018 scheme, and A.P. is grateful for a British Council, UK-Israel Science Fellowship, 2018–2019 scheme. During writing and revision, E.Z. has received funding from the MINDED fellowship of the European Union's Horizon 2020 research and innovation program under the Marie Skłodowska-Curie grant agreement No. 754490. Authors thank members of the E.G. and D.S research groups. Authors are grateful to Prof. Annalisa Pastore for valuable comments on this manuscript.

## Author contributions
A.P., E.Z., and D.S. planned and designed the experiments. E.Z. and D.S. developed the original idea. A.P performed of the reported experiments and data analysis. E.Z., M.F.P., D.E.A., and G.M. performed preliminary in vitro experiments. The manuscript was mainly drafted by A.P. and E.Z. with D.S and E.G. All authors read and approved the manuscript for publication.

## Competing interests
The authors declare no competing interests.
