## [Peer Review File · Communications Biology]

Reviewers' comments:

Reviewer #1 (Remarks to the Author):

This study provides a rational design strategy to develop dual inhibitors against amyloid aggregation of both A β 42 and hIAPP, additionally with disaggregation ability. Apart from the detailed experimental results, there are some issues need to be addressed before publication.

1. There exists some unclear descriptions and inconsistent results from experiments.

* while the authors claimed that "No significant variation on the A β 42 secondary structure was observed in the presence of the control WW", the CD spectrum of A β 42: WW (1:5) actually exhibited an alpha-helix profile rather than the β -sheet profile displayed by A β 42. How to explain that differences and the effect of WW on A β 42 structural transition?

* ThT, CD and TEM data indicate no inhibitory effect of WW on hIAPP aggregation, but no Congo red-related birefringence was observed in the hIAPP/WW fibrils. How to explain such inconsistency.

* Also, the authors ignored the differences in the fibril morphology of hIAPP and hIAPP:WW (1:5) in Figure 3.

2. While the authors provided many characterization results for the aggregation in the presence and absence of different designed molecules, no detailed comparison and analysis were discussed throughout the manuscript.

* The fibril contents of both pre-formed A β 42 and hIAPP fibrils increased upon treated with WW in contrast to the inhibition from WGal and its derivatives, but the resulted cytotoxicity also decreased as exhibited in Fig.9. How to compare and explain these results?

3. The study aimed to increase solubility and target selectivity of the hybrid molecule by conjugating tryptophan to a glycan unit and its derivatives. However, no fair comparison were made and emphasized on these three tryptophan-galactosylamine hybrids (WGal, WGalNH₂, WGalNAc). Also the mechanistic role of galactose and its different derivatives have not been analyzed throughout this manuscript.

Overall, the manuscript can be improved significantly by providing in-depth discussion and comprehensive data comparison and analysis.

Reviewer #2 (Remarks to the Author):

Brief summary of the manuscript:

In the paper titled "Tryptophan-galactosylamine conjugates inhibit and disaggregate amyloid fibrils of A β 42 and hIAPP peptides while reducing their cytotoxicity" the authors describe the use of the aromatic amino acid, tryptophan, to disrupt amyloid formation of the peptides Abeta42 and hIAPP. They improved upon the solubility of tryptophan through the creation of new derivatives with galacosylamine hybrids. The authors postulate that these hybrids, WGal, WGalNH₂, and WGalNAc, where WW serves as a negative control, are effective inhibitors of Abeta42 and hIAPP aggregation through the analysis of aggregation kinetics (ThT), circular dichroism (CD), electron microscopy (EM), and congo red birefringence (CR). Similarly, the authors use these hybrids in experiments of disaggregation of Abeta42 and hIAPP. Finally, the authors test the toxicity of hybrids then, in the presence of either amyloid, show the effect of hybrids on amyloid-induced cytotoxicity.

Overall impression of the work

Amyloid diseases remain a major problem in human health and disease. More than 99% of therapeutics in clinical trials in the past decade failing to modify Alzheimer's Disease progression, just one of many amyloid related diseases. Despite recent failures in clinical trials aimed at disrupting amyloidogenic peptide formation and aggregation, the research community should not halt discovery research in this area. I appreciate the authors making a new contribution here. Indeed, non-toxic small molecule disruptors of this pathway may be more optimal than antibody-based clearance mechanisms.

This manuscript presents an interesting hypothesis that adds to the growing literature that some natural products may disrupt amyloidogenesis. That is that a highly soluble hybrid of an aromatic tryptophan may intercalate in the growing fibril and either disrupt further aggregation or even break preexisting fibrils. The experimental evidence presented in the results section is intriguing, however, I would appreciate additional experiments to support their claims.

Specific comments, with recommendations for addressing each comment

Inhibition of Aβ42 and hIAPP aggregation as measured by ThT is intriguing. The authors show a dose dependent effect for WGal, WGalNH₂, and WGalNac, but not WW. CD data also show a dose dependent conformational changes in the presence of hybrids. To add more value to the understanding of this process, the authors should employ the AmyloFit tool (PMID: 26741409, PMID: 25686087). With an expanded inhibitor dose range, this tool models changes in the curve and provides valuable insight into the mechanism of action of the inhibitor. It would be interesting and valuable to understand how these tryptophan hybrid compounds disrupt aggregation kinetics (primary nucleation, secondary nucleation, fibril elongation).

EM and CR work is intriguing, though can be highly subjective. The authors should quantify a large collection of images to make a convincing case that the fibril mass is indeed diminished, and perhaps remaining fibrils are shorter in length. The authors should determine if the remaining fibrils after inhibitor treatment retain CR birefringence. Furthermore, it would be valuable to determine what species remain after inhibition of disaggregation experiments. Combining ultracentrifugation (remove fibrils), size exclusion chromatography (determine oligomers versus dimer-monomer), and quantification of each population would aid in determining a hypothesis on the mechanism of action for these tryptophan hybrid molecules.

The cytotoxicity data are intriguing but could be more complete. A full dose curve would add value and the authors could present an inhibitory dose concentration. I am confused by the effective cytotoxicity inhibition of the control, WW, because the ThT, CD, EM, and CR data would indicate the persistence of fibrils for both Aβ42 and hIAPP. Therefore, if the WW cytotoxicity data are accurate, the alternative hypothesis would then be that the mechanism of action would be fibril detoxification. This therefore should be explored and WW should not be treated as a negative control. There was limited discussion of this phenomenon in the text.

Finally, I would appreciate more discussion of inhibitors of amyloidogenesis in the discussion, especially in light of the recent clinical trial failures. Small molecule inhibitors could be an important step forward, if successful, however clinical application would remain a major hurdle.

Response to the referees' comments

We would like to thank all the referees and the editorial team for their careful evaluation and valuable comments and suggestions. We have addressed all of the points mentioned by them. We hope that you will find this revised version suitable for publication in Communications Biology.

To the editorial team: to offer a clear point-by-point discussion, in some cases we have split the reviewers' comments into its different components. In this document, our comments are in bold and the alterations and insertions of new text are highlighted in yellow in the revised manuscript.

Referee expertise:

Referee #1: peptide engineering

Referee #2: chemical biology

Reviewers' comments:

Reviewer #1 (Remarks to the Author):

This study provides a rational design strategy to develop dual inhibitors against amyloid aggregation of both A β 42 and hIAPP, additionally with disaggregation ability. Apart from the detailed experimental results, there are some issues need to be addressed before publication.

1. There exists some unclear descriptions and inconsistent results from experiments.

1a. While the authors claimed that “No significant variation on the A β 42 secondary structure was observed in the presence of the control WW”, the CD spectrum of A β 42: WW (1:5) actually exhibited an alpha-helix profile rather than the β -sheet profile displayed by A β 42. How to explain that differences and the effect of WW on A β 42 structural transition?

Author's response

First of all, we would like to thank the reviewer for careful evaluation, appreciation and valuable comments.

The reviewer is indeed correct in his/her observation and has identified a mistake in Figure 1 that we have not previously spotted. By genuine mistake, we selected incorrectly plotted CD curves to represent the effect of WW on the secondary structure of A β 42 (Figure 1c) and hIAPP (Figure 1f). We have rectified this error in the revised manuscript, which now contains the correct image clearly showing that the CD spectrum of A β 42 and hIAPP in the presence of 5-fold molar excess of WW is consistent with a β -sheet rich conformation. Indeed, it appears that, if anything, WW may increase the β -sheet content of both peptides. The revised spectra are show below:

Figure 1. Inhibition of amyloid formation of A β 42 and hIAPP by the tryptophan-galactosylamine hybrid molecules. (a) ThT kinetics of A β 42 aggregation in the absence or presence of 5-fold molar excess of the tested molecules. (b) Dose-dependent inhibition of A β 42 aggregation by the tested molecules. (c) CD analysis of conformational changes of the A β 42 aggregates in the absence or presence of the tested molecules. (d) ThT kinetics of hIAPP aggregation in the absence or presence of 5-fold molar excess of the tested molecules. (e) Dose-

dependent inhibition of hIAPP aggregation by the tested molecules. (f) CD analysis of conformational changes of the hIAPP aggregates in the absence or presence of the tested molecules.

1b. ThT, CD and TEM data indicate no inhibitory effect of WW on hIAPP aggregation, but no Congo red-related birefringence was observed in the hIAPP/WW fibrils. How to explain such inconsistency.

Author's response

The Congo red birefringence analysis of hIAPP in the presence of 5-fold of WW showed green-gold birefringence, in agreement with the ThT, CD and TEM results. The birefringence is clearly noticeable in the image. We believe that the error here was in the text, in which we stated that “no Congo red-related birefringence was detected”. We apologise to the reviewer; this was obviously a typo. The mistake has been rectified in the revised manuscript. The revised text is reported below (main text, lines 195-198):

“In contrast, in the presence of the control WW, hIAPP amyloid fibrils were clearly visible in a more densely packed network and a clear gold-green birefringence was detectable. These results may suggest that not only WW was unable to inhibit the aggregation of hIAPP, but also that this dipeptide may promote further aggregation of hIAPP.”

1c. Also, the authors ignored the differences in the fibril morphology of hIAPP and hIAPP:WW (1:5) in Figure 3.

Author's response

The reviewer is correct. In the paper we have focused only on the mere presence/absence of aggregates. As the reviewer pointed out, the aggregates formed by hIAPP in the presence of WW have a more bundle like and dense fibrillar appearance. This may be an indication that these aggregates still preserve the same amyloid core, but they appear more densely packed when incubated with WW. All these observations have now been given more space in the Results and Discussion sections (main text, lines 195-196).

“In contrast, in the presence of the control WW, hIAPP amyloid fibrils were clearly visible in a more densely packed network”.

2. While the authors provided many characterization results for the aggregation in the presence and absence of different designed molecules, no detailed comparison and analysis were discussed throughout the manuscript.

Author’s response

The reviewer has pointed out correctly, we have not discussed a detail comparison between the effects of the different hybrid molecules. This is because in the majority of our results the effects of all hybrid molecules on aggregation behaviour and secondary structure of our target peptides were comparable. Following the suggestion of the Reviewer, we thought of calculating the *p*-values of the reduction of the aggregation caused by every hybrid molecule compared to the others. We found that, albeit the differences in the inhibitory effect among the three molecules are not pronounced, there is indeed a statistically significant trend throughout our results suggesting that WGalNAc may be a slightly more efficient aggregation inhibitor. We have now discussed these results in the revised manuscript and added all the new statistical data in the revised Supporting Information (Figures S2 and S9).

The new text in the Discussion section of the revised manuscript (lines 412-421) states as follows: “A closer look into the results of the ThT-binding assays suggests that although the differences between the effects of the inhibitors are small, WGalNAc is on average a significantly stronger amyloid inhibitor compared to WGal and WGalNH₂, as indicated by the *p*-value calculations (Figures S2 and S9). The superiority of WGalNAc may be accounted for by its acetyl group. This moiety may actively contribute to increase the energetically favourable CH- π stacking between the cluster of carbon atoms of the hexose ring and the target peptides^{107,108}. Moreover, the larger molecular radius of –NAc compared to –NH₂ and –OH may allow WGalNAc to interact more stably with a higher number of atoms of the target peptides¹⁰⁹. Finally, the slightly more hydrophobic acetyl group may also participate in binding the amyloidogenic peptides tested, and lead the whole hybrid molecule WGalNAc towards a more balanced equilibrium between the ideal hydrophilicity and hydrophobicity¹¹⁰.”

The new figures added to the Supporting Informations are also reported below:

Figure S2. Percentage of amyloid formation and statistical analysis of the effect of the tested molecules on aggregation of Aβ42 and hIAPP. (a) Statistical significance and *p*-values of the inhibitory effect of the hybrid molecules compared to Aβ42 alone and compared to each other; (b) Statistical significance and *p*-values of the inhibitory effect of the hybrid molecules compared to hIAPP alone and compared to each other. The percentage of amyloid formation was calculated the end of the ThT-binding assays. N.S.: not significant; *: *p*-value < 0.05; **: *p*-value < 0.005; ***: *p*-value < 0.001.

Figure S9. Percentage of amyloid disruption and statistical analysis of the effect of the tested molecules on the disruption of pre-formed Aβ42 and hIAPP aggregates. (a) Statistical significance and *p*-values of Aβ42 aggregate disruption by the hybrid molecules compared to Aβ42 alone and compared to each other; (b) Statistical significance and *p*-values of hIAPP aggregate disruption by the hybrid molecules compared to hIAPP alone and compared to each other. The percentage of pre-formed amyloid disruption was calculated at the end of the ThT-binding assays. N.S.: not significant; *: *p*-value < 0.05; **: *p*-value < 0.005; *: *p*-value < 0.001.**

2. The fibril contents of both pre-formed A β 42 and hIAPP fibrils increased upon treated with WW in contrast to the inhibition from WGal and its derivatives, but the resulted cytotoxicity also decreased as exhibited in Fig.9. How to compare and explain these results?

Author's response

Thanks to the reviewer for bringing up this point. The effect of WW on the aggregation of our target peptides is indeed interesting and we have mentioned this in already in the Discussion section of the original manuscript. We are more than happy to discuss this intriguing aspect further, and we have done so in the revised version of the manuscript and included our hypothesis on how the increment of amyloidogenic species upon incubation with WW may result in less toxic aggregates. The revised text that has been added in the Discussion section of the revised manuscript (lines 422-437) and the new Figure 9 are provided below:

“Alongside the designed hybrid molecules, all experiments performed in this work included the dipeptide WW, here referred to as “negative control” which allowed to assert the effect of the galactosylamine derivatives in comparison to the tryptophan alone. However, in some circumstances, WW displayed a stimulatory effect on the aggregation of A β 42 and hIAPP. This behaviour became more pronounced and statistically significant (**Figure S2** and **S9**) during the disaggregation of pre-formed fibrils (**Figures 4-6**). These results could be explained by the amyloidogenic nature of tryptophan and by its ability to self-assemble into fibrils, resulting in an additive aggregation level^{55,80}. The co-presence of aggregating peptides may have functioned as further nucleation agent for the π - π stacking of tryptophan, which, on its end, could also have consequently induced further A β 42 and hIAPP aggregation into mature amyloid fibrils. This effect, however, did not translate into an increment of toxicity in the in-cell assays: albeit not as efficiently, WW, like the hybrid molecules, reduced the cytotoxicity induced by extra-cellular pre-aggregated A β 42 and hIAPP. To explain the partial inhibitory effect of WW on A β 42 and hIAPP-induced toxicity, we postulate that WW may have driven the aggregates of these peptides towards larger bundle of fibrils, which are reported to be less toxic than smaller oligomeric forms¹¹¹⁻¹¹⁴. WW's effect on cytotoxicity (**Figures 9**), however statistically significant, is very weak when compared to the hybrid molecules: EC₅₀ and E_{max} of all hybrids are always between 2.5 and 5-fold lower than the ones calculated for WW.”

Figure 9. Effects of the tryptophan-galactosylamine hybrid molecules towards cytotoxicity of pre-formed amyloids. (a) Aβ42- and (b) hIAPP-induced cytotoxicity in the presence or absence of various doses of the tested molecules were evaluated using XTT reduction assay. EC₅₀ and E_{max} of the tested molecules when applied to reduce the cytotoxicity induced by pre-formed fibrils of (c) Aβ42 and (d) hIAPP. ####p < 0.001, compared to the untreated cells. **p < 0.05, *p < 0.005 and ***p < 0.001 compared to Aβ42 (a) and hIAPP (b) treatment. Untreated cells were considered as 100% viable.

3. The study aimed to increase solubility and target selectivity of the hybrid molecule by conjugating tryptophan to a glycan unit and its derivatives. However, no fair comparison were made and emphasized on these three tryptophan-galactosylamine hybrids (WGal, WGalNH₂, WGalNAc). Also the mechanistic role of galactose and its different derivatives have not been analyzed throughout this manuscript.

Overall, the manuscript can be improved significantly by providing in-depth discussion and comprehensive data comparison and analysis.

Author's response

The reviewer has indeed spotted a weak point in our discussion. Thank you for the suggestion. We have now included several more observations and a more critical data analysis in our Discussion. We have also added two new figures in our Supporting Information (Figures S2 and S9) in which we illustrate the *p*-value calculations among the effect of all hybrid molecules. Please, refer to the answer to point 2 for the text we now added to the Discussion regarding the comparison between the hybrid molecules (lines 422-437) and for examining the newly added Figures in the Supporting Information.

The Discussion section now also includes our hypothesis on the possible mechanistic effect of the inhibitors (lines 388-411). We report it below:

“At the core of the amyloidogenic process there are conformational changes of the native, soluble peptides into unstable oligomers that subsequently assemble into insoluble protofibrils and fibrils⁹⁸. The formation of oligomers in the absence of other pre-existent larger peptide species is referred to as primary nucleation⁹⁹. These oligomers can grow into fibers by the addition of other oligomers (elongation) or monomers (secondary nucleation)⁶⁰. The kinetics of amyloid formation generally follows a sigmoidal trend, in which a stable lag phase, an exponential phase and a plateau, the stage by which no more monomers are available, can be distinguished^{100,101}. According to the mechanism of action of a potential inhibitor, any of the phases of the aggregation process may be effected. All our *in vitro* analyses demonstrated an appreciable amyloid inhibitory effect on both A β 42 and hIAPP by all the tryptophan-galactosylamine hybrids. The data collected for WGal, WGalNH₂ and GalNAc show that there is no kinetic effect caused by the inhibitors on primary nucleation or elongation (lag phase or exponential phase). Instead, the plateau levels of both A β 42 and hIAPP are significantly reduced by the presence of the hybrid molecules. To speculate on the mechanistic role that these inhibitors are playing on the aggregation of the target peptides, it is important to bear in mind that there are many ways by which small molecules may model the aggregation pathway^{102,103}. Among these, it has been reported that various inhibitors reduce the amount of amyloids by redirecting the aggregation towards alternative off-path products, rather than preventing the self-assembly of the monomers^{41,104–106}. We believe that this is the case for the hybrid molecules studies here: the lack of an obvious kinetic effect on primary/secondary nucleation or elongation suggests the sequestration of the monomers and oligomers to a parallel alternative pathway that does not lead to amyloid

formation but to smaller and often amorphous assemblies. This hypothesis is supported by the ca. 50% reduction of ThT-positive aggregates in the presence of the inhibitors; by the variation in A β 42 and hIAPP's secondary structures; by the lack of fiber-like assemblies as shown by TEM and Congo red birefringence; and by the significant lower cellular toxicity.”

Reviewer #2 (Remarks to the Author):

Brief summary of the manuscript: In the paper titled “Tryptophan-galactosylamine conjugates inhibit and disaggregate amyloid fibrils of A β 42 and hIAPP peptides while reducing their cytotoxicity” the authors describe the use of the aromatic amino acid, tryptophan, to disrupt amyloid formation of the peptides Abeta42 and hIAPP. They improved upon the solubility of tryptophan through the creation of new derivatives with galactosylamine hybrids. The authors postulate that these hybrids, WGal, WGalNH₂, and WGalNAc, where WW serves as a negative control, are effective inhibitors of Abeta42 and hIAPP aggregation through the analysis of aggregation kinetics (ThT), circular dichroism (CD), electron microscopy (EM), and congo red birefringence (CR). Similarly, the authors use these hybrids in experiments of disaggregation of Abeta42 and hIAPP. Finally, the authors test the toxicity of hybrids then, in the presence of either amyloid, show the effect of hybrids on amyloid-induced cytotoxicity.

Overall impression of the work: Amyloid diseases remain a major problem in human health and disease. More than 99% of therapeutics in clinical trials in the past decade failing to modify Alzheimer’s Disease progression, just one of many amyloid related diseases. Despite recent failures in clinical trials aimed at disrupting amyloidogenic peptide formation and aggregation, the research community should not halt discovery research in this area. I appreciate the authors making a new contribution here. Indeed, non-toxic small molecule disruptors of this pathway may be more optimal than antibody-based clearance mechanisms.

This manuscript presents an interesting hypothesis that adds to the growing literature that some natural products may disrupt amyloidogenesis. That is that a highly soluble hybrid of an aromatic tryptophan may intercalate in the growing fibril and either disrupt further aggregation or even break preexisting fibrils. The experimental evidence presented in the results section is intriguing, however, I would appreciate additional experiments to support their claims.

Specific comments, with recommendations for addressing each comment:

1a. Inhibition of Abeta42 and hIAPP aggregation as measured by ThT is intriguing. The authors show a dose dependent effect for WGal, WGalNH₂, and WGalNAc, but not WW. CD data also show a dose dependent conformational changes in the presence of hybrids. To add more value to the understanding of this process, the authors should employ the AmyloFit tool (PMID:

26741409, PMID: 25686087). With an expanded inhibitor dose range, this tool models changes in the curve and provides valuable insight into the mechanism of action of the inhibitor. It would be interesting and valuable to understand how these tryptophan hybrid compounds disrupt aggregation kinetics (primary nucleation, secondary nucleation, fibril elongation).

Author's response

We appreciate the Reviewer's careful evaluation and valuable suggestions. We would also like to thank the Reviewer for suggesting AmyloFit.

In all our ThT-binding assays, we observed a variation in the final amount of ThT-positive aggregates, represented by lower plateau levels. Often, lower plateau levels are associated with a kinetic effect of the inhibitors, that bind either the monomers, resulting in longer lag phases, or the oligomers, resulting in a slower elongation phases (Trends in Pharmacological Sciences 2014, 35, 3). However, in our case, the inhibitors do not appear to alter neither lag phase nor elongation phase. Thanks to the suggestion of the Reviewer, we have attempted to engage with AmyloFit to understand better this conflicting evidence. When we tried using our data of A β 42/hIAPP in the absence of the inhibitors, the software could fit very well the aggregation curve according to the model "secondary nucleation dominated, unseeded" (please, refer to the kinetics parameters provided below in the figures and tables). However, AmyloFit did not manage to fit well the curve in the presence of the inhibitors under the same set parameters. Instead, it generated fitting curves in which both primary and secondary nucleation processes are affected, together with lower plateau levels. After talking directly with Dr. Georg Meisl, one of the creators of the AmyloFit, we have concluded that, taken together, the misfit of the curve and the difference in plateau levels show that the inhibitors are probably initiating a parallel process at later stages of the aggregation curve, rather than effecting the kinetics of early stages. Since kinetics are not affected, unfortunately AmyloFit cannot offer further information.

We have added some speculations on the potential mechanistic effect of the inhibitor in the revised Discussion section (lines 388-411). The new text states as follows:

"At the core of the amyloidogenic process there are conformational changes of the native, soluble peptides into unstable oligomers that subsequently assemble into insoluble protofibrils

and fibrils⁹⁸. The formation of oligomers in the absence of other pre-existent larger peptide species is referred to as primary nucleation⁹⁹. These oligomers can grow into fibers by the addition of other oligomers (elongation) or monomers (secondary nucleation)⁶⁰. The kinetics of amyloid formation generally follows a sigmoidal trend, in which a stable lag phase, an exponential phase and a plateau, the stage by which no more monomers are available, can be distinguished^{100,101}. According to the mechanism of action of a potential inhibitor, any of the phases of the aggregation process may be effected. All our *in vitro* analyses demonstrated an appreciable amyloid inhibitory effect on both A β 42 and hIAPP by all the tryptophan-galactosylamine hybrids. The data collected for WGal, WGalNH₂ and GalNAc show that there is no kinetic effect caused by the inhibitors on primary nucleation or elongation (lag phase or exponential phase). Instead, the plateau levels of both A β 42 and hIAPP are significantly reduced by the presence of the hybrid molecules. To speculate on the mechanistic role that these inhibitors are playing on the aggregation of the target peptides, it is important to bear in mind that there are many ways by which small molecules may model the aggregation pathway^{102,103}. Among these, it has been reported that various inhibitors reduce the amount of amyloids by redirecting the aggregation towards alternative off-path products, rather than preventing the self-assembly of the monomers^{41,104-106}. We believe that this is the case for the hybrid molecules studies here: the lack of an obvious kinetic effect on primary/secondary nucleation or elongation suggests the sequestration of the monomers and oligomers to a parallel alternative pathway that does not lead to amyloid formation but to smaller and often amorphous assemblies. This hypothesis is supported by the ca. 50% reduction of ThT-positive aggregates in the presence of the inhibitors; by the variation in A β 42 and hIAPP's secondary structures; by the lack of fiber-like assemblies as shown by TEM and Congo red birefringence; and by the significant lower cellular toxicity.”

An example of AmyloFit applied to Aβ42 and to one of the hybrid molecules.

SUMMARY OF FITTING RESULTS					
Model used:	Secondary_Nucleation_Dominated_noseed				
Mean squared error Aβ42:	2.40E-13	=sum[(y _i - f(x _i)) ²]/(N_datapoints - N_free_parameters)			
Mean squared error Aβ42+WGalNH ₂ :	7.28E-13				
Parameters from fit:					
Dataset	n _c	m ₀	kptimesk2 (k ₊ k ₂)	n ₂	kptimeskn (k ₊ k _n)
Units	unitless	mol * l ⁻¹	conc ^{-n₂-1} time ^{-2}	unitless	conc ^{-n_c} time ^{-2}
Abeta	2	1.00E-05	1.29E+15	2	207697687
Aβ42+WGalNH ₂	2	6.64E-06	1.29E+15	2	207697687

AmyloFit kinetics results for Aβ42 and one of the hybrid molecules.

An example of AmyloFit applied to hIAPP and to one of the hybrid molecules.

SUMMARY OF FITTING RESULTS					
Model used:	Secondary_Nucleation_Dominated_noseed				
Mean squared error IAPP:	3.12E-14	=sum[(y _i - f(x _i)) ²]/(N_datapoints - N_free_parameters)			
Mean squared error IAPP+WGalNH ₂ :	3.86E-13				
Parameters from fit:					
Dataset	n _c	m ₀	k _p timesk ₂ (k ₊ k ₂)	n ₂	k _p timesk _n (k ₊ k _n)
Units	unitless		conc ^{-n₂-1} time ^{-2}	unitless	conc ^{-n_c} time ^{-2}
Aβ ₄₂	2	1.00E-05	8.44E+14	2	4.71E+08
Aβ ₄₂ +WGalNH ₂	2	4.08E-06	8.44E+14	2	4.71E+08

AmyloFit kinetics results for hIAPP and one of the hybrid molecules.

Where:

k_n: primary nucleation rate constant

k₊: elongation rate constant

n_c: primary nucleus size

m₀: concentration

k₂: secondary nucleation rate constant

1b. EM and CR work is intriguing, though can be highly subjective. The authors should quantify a large collection of images to make a convincing case that the fibril mass is indeed diminished, and perhaps remaining fibrils are shorter in length. The authors should determine if the remaining fibrils after inhibitor treatment retain CR birefringence.

“This is what I mean: The Congo red (CR) birefringence are interesting and helpful in understanding the classic definition of amyloid in their system, however they are single point data. Compared to control, I as the reader would like to see a quantified analysis. It appears to me that the WW (negative control treatment) had a significant effect compared to peptide alone for both A β 42 and hIAPP. It is unclear, because a lack of quantification, if the image of WW is representative or not, though, of course, I would assume it is. If it is, then perhaps the effect of WW in some, but not all figures makes some sense.”

Author’s response

We thank the reviewer for the observations and for the clarification offered. We are aware that the Congo red images by themselves may not be sufficient to evaluate the effect of the inhibitors on peptide aggregation. We have now added further images to the Supporting Information (Figures S6, S7, S13 and S14) to convince the reader that WW does not have any significant effect on the amyloid formation of either peptide while all other WGal derivatives do, in agreement with the quantifiable ThT assays and with CD and TEM results. We also appreciate that, as the reviewer pointed out, we may have overlooked a possible effect of WW. To verify this, we applied the student *t*-test at the end of the aggregation curve and calculate the *p*-value between A β 42/hIAPP alone and in the presence of WW (Figure S2). According to the *p*-values, the effect of WW on the aggregation of A β 42/hIAPP is not significant at any of the employed concentration. However, a significant pejorative effect of WW was detected for the disaggregation assays on both peptides (Figure S9). Please see our answer to the comment 2a of Reviewer 1, in which we address the possible reasons for a pejorative effect of WW on the aggregation of the two peptides under investigation. We have also given more space to the discussion of this aspect in the revised version of the manuscript and added some statistical analysis on the effect of WW in the Supporting Information (Figure S2 and S9).

We are also grateful for the reviewer’s input on Congo red birefringence but we would like to remark on certain limitations of this technique. Congo red birefringence is usually

employed as a qualitative technique. It allows to appreciate apparent morphological differences (when present) and to establish the overall reduction/increment in the level of aggregation (when clearly visible). However, employing the acquired images for quantification of birefringence intensity and comparison of intensity levels is very difficult. Moreover, in the concentrations and time-frame used in our experiments, it is near to impossible to isolate single fibrils in a number that may render the measurement of their length significant. We can see that the fibrils appear shorter, but, under our experimental conditions, we are in no position to determine their exact length with accuracy. If we were to attempt to quantify the amount of aggregates base on these images, we would be making a considerable amount of assumptions. For these reasons, our birefringence images are always accompanied by TEM micrographs, which instead reveal a factual picture of the real aggregate morphology, and by ThT and CD analysis, which give a quantitative estimation of aggregate amount and conformation.

More Congo red-stained images have now been added for clarification to the revised Supporting Information (Figures S6, S7, S13 and S14). We have provided these images also below:

Figure S2. Percentage of amyloid formation and statistical analysis of the effect of the tested molecules on aggregation of Aβ42 and hIAPP. (a) Statistical significance and *p*-values of the inhibitory effect of the hybrid molecules compared to Aβ42 alone and compared to each other; (b) Statistical significance and *p*-values of the inhibitory effect of the hybrid molecules compared to hIAPP alone and compared to each other. The percentage of amyloid formation was calculated the end of the ThT-binding assays. N.S.: not significant; *: *p*-value < 0.05; **: *p*-value < 0.005; ***: *p*-value < 0.001.

Figure S9. Percentage of amyloid disruption and statistical analysis of the effect of the tested molecules on the disruption of pre-formed Aβ42 and hIAPP aggregates. (a) Statistical significance and *p*-values of Aβ42 aggregate disruption by the hybrid molecules compared to Aβ42 alone and compared to each other; (b) Statistical significance and *p*-values of hIAPP aggregate disruption by the hybrid molecules compared to hIAPP alone and compared to each other. The percentage of pre-formed amyloid disruption was calculated at the end of the ThT-binding assays. N.S.: not significant; *: *p*-value < 0.05; **: *p*-value < 0.005; *: *p*-value < 0.001.**

Figure S6. Analysis of A β 42 fibrils in the absence or presence of the tryptophan-galactosylamine hybrid molecules. Additional Congo red stained birefringence representative images of A β 42 fibrils in the absence or presence of the 5-fold molar excess of the tested molecules. All scale bars, 100 μ m.

Figure S7. Analysis of hIAPP fibrils in the absence or presence of the tryptophan-galactosylamine hybrid molecules. Additional Congo red stained birefringence representative images of hIAPP fibrils in the absence or presence of the 5-fold molar excess of the tested molecules. All scale bars, 100 μm .

Figure S13. Analysis of pre-formed A β 42 fibrils in the absence or presence of the tryptophan-galactosylamine hybrids. Additional Congo red stained birefringence representative images of A β 42 fibrils in the absence or presence of the 5-fold molar excess of the tested molecules. All scale bars, 100 μ m.

Figure S14. Analysis of pre-formed hIAPP fibrils in the absence or presence of the tryptophan-galactosylamine hybrids. Additional Congo red stained birefringence representative images of hIAPP fibrils in the absence or presence of the 5-fold molar excess of the tested molecules. All scale bars, 100 μm.

1c. Furthermore, it would be valuable to determine what species remain after inhibition of disaggregation experiments. Combining ultracentrifugation (remove fibrils), size exclusion chromatography (determine oligomers versus dimer-monomer), and quantification of each population would aid in determining a hypothesis on the mechanism of action for these tryptophan hybrid molecules.

Author's response

We are certain that the reviewer is aware that aggregation is an extremely dynamic process in which multiple states co-exist and the balance is constantly changing. To pinpoint the exact amount of each oligomeric specie present at any given point during protein aggregation is a very difficult task. As the reviewer suggests, it would indeed be very interesting to have further information on mechanistic aspects of the effect of these hybrid molecules on the peptide aggregation. The fact that the addition of the hybrid molecules to pre-formed aggregates causes a strong reduction of ThT-associated fluorescence is a sign that they act on large oligomers/fibres, since at that state of apparent completed aggregation only an infinitesimal portion of monomers/small oligomers is present. We have expressed further thoughts on this in the answer to point 1a, together with the output of AmyloFit.

To answer this question experimentally, the combination of AUC and SEC in principle could be a splendid approach, but practically there are many limitations. We have tried several times to follow A β 42 aggregation via SEC starting from t0 to 24 hours into its aggregation (below some images to show our attempts), but, in our hands, this was always unsuccessful, irrespective of the protein concentration or conditions used. We share here an example with the reviewers, together with the calibration data of the column used and the presumed isolated A β 42 oligomers:

Column resolution as per calibration			Aβ42 oligomers resolution		
Protein	Elution vol. (ml)	MW (KDa)	Elution vol. (ml)	Presumed MW (KDa)	Oligomers
Conalbumin	11.8	75	9.2	169.3	37.5
Ovalbumin	12.7	44	10.4	111.0	24.6
Carbonic anhydrase	14.2	29	15.4	19.1	4.2
Cytochrome C	16.6	12.3	16.6	12.5	2.8
Aprotinin	18.6	6.5	19.3	4.8	1.1
Vitamin B ₁₂	23	1.3	20.6	3.1	Noise

The figure we provided shows the chromatograms of 20 μM Aβ42 in PBS injected at different time points onto a 30 ml Superdex75 column. As clearly evident, no change in the chromatogram is visible between time 0 and 4 hours since the start of the aggregation process, which instead was clearly detectable by ThT-binding assay and CD spectroscopy. We believe that there are many factors that concur to the failure of this method. Among these, we think the most relevant in our case are:

- the properties of the aggregation phenomenon, which is a constant and dynamic process that continues also during data acquisition;

- the limit of the cut-off of the chosen columns, none of which can be suitable enough to cover the whole possible range of different molecular weights;
- the large variety of oligomeric species present at the same time, which means that none of them will be particularly well defined as a absorbance peak during elution, but that will be represented by overlapping peaks, requiring a very high protein concentration to visualise them.

For these reasons, and considering our previous attempt to tackle the issue raised by the reviewer, we believe that, in our case, this experimental approach may be technically impossible.

2a. The cytotoxicity data are intriguing but could be more complete. A full dose curve would add value and the authors could present an inhibitory dose concentration.

Author's response

We appreciate the remark and agree that a full dose-response curve would render our in-cell results more elegant and complete. We have now performed the additional necessary experiments and are able to compare apparent EC_{50} and E_{max} for all the tested molecules. We have included these results in the main text of the manuscript (Figure 9) and in the Supporting Information (Figure S15). These new figures are also provided below:

Figure 9. Effects of the tryptophan-galactosylamine hybrid molecules towards cytotoxicity of pre-formed amyloids. (a) A β 42- and (b) hIAPP-induced cytotoxicity in the presence or absence of various doses of the tested molecules were evaluated using XTT reduction assay. EC₅₀ and E_{max} of the tested molecules when applied to reduce the cytotoxicity induced by pre-formed fibrils of (c) A β 42 and (d) hIAPP. ###p < 0.001, compared to the untreated cells. **p < 0.05, *p < 0.005 and ***p < 0.001 compared to A β 42 (a) and hIAPP (b) treatment. Untreated cells were considered as 100% viable.

Figure S15. Dose-response curves of cell viability increment of calculated from XTT assay (Figure 9). (a) Increment cell viability (%) of SH-SY5Y upon treatment of different concentrations of the tested molecules on A β 42 (13 μ M) preformed fibrils and (b) Increment cell viability (%) of HEK-293 cells upon treatment of different concentrations of the tested molecules on hIAPP (10 μ M) preformed fibrils.

2b. I am confused by the effective cytotoxicity inhibition of the control, WW, because the ThT, CD, EM, and CR data would indicate the persistence of fibrils for both A β 42 and hIAPP. Therefore, if the WW cytotoxicity data are accurate, the alternative hypothesis would then be that the mechanism of action would be fibril detoxification. This therefore should be explored and WW should not be treated as a negative control. There was limited discussion of this phenomenon in the text.

Author's response

The reviewer is raising a very important issue, since the restorative effect of WW on A β 42/IAPP-induced cytotoxicity has intrigued us as well. Since we could only speculate, at the moment of submission we decided not to include this part in the Discussion, but the reviewer is perfectly right in thinking that it should instead deserve some explanation. We now propose a thesis that would explain the behaviour of WW in the Discussion section (lines 422-437). The text is also reported below:

“Alongside the designed hybrid molecules, all experiments performed in this work included the dipeptide WW, here referred to as “negative control” which allowed to assert the effect of the galactosylamine derivatives in comparison to the tryptophan alone. However, in some circumstances, WW displayed a stimulatory effect on the aggregation of A β 42 and hIAPP. This behaviour became more pronounced and statistically significant (**Figure S2** and **S9**) during the disaggregation of pre-formed fibrils (**Figures 4-6**). These results could be explained by the amyloidogenic nature of tryptophan and by its ability to self-assemble into fibrils, resulting in an additive aggregation level^{55,80}. The co-presence of aggregating peptides may have functioned as further nucleation agent for the π - π stacking of tryptophan, which, on its end, could also have consequently induced further A β 42 and hIAPP aggregation into mature amyloid fibrils. This effect, however, did not translate into an increment of toxicity in the in-cell assays: albeit not as efficiently, WW, like the hybrid molecules, reduced the cytotoxicity induced by extra-cellular pre-aggregated A β 42 and hIAPP. To explain the partial inhibitory effect of WW on A β 42 and hIAPP-induced toxicity, we postulate that WW may have driven the aggregates of these peptides towards larger bundle of fibrils, which are reported to be less toxic than smaller oligomeric forms¹¹¹⁻¹¹⁴. WW's effect on cytotoxicity, however statistically significant, is very weak when

compared to the hybrid molecules: EC_{50} and E_{max} of all hybrids are always between 2.5 and 5-fold lower than the ones calculated for WW.”

3. Finally, I would appreciate more discussion of inhibitors of amyloidogenesis in the discussion, especially in light of the recent clinical trial failures. Small molecule inhibitors could be an important step forward, if successful, however clinical application would remain a major hurdle.

Author’s response

We have added new text highlighted in yellow as Discussion in the later part of revised manuscript as suggested by the reviewer (lines 438-454). The text is also reported below:

“Significant breakthroughs in inhibition of disease associated amyloid aggregation have led to different therapeutic strategies, which include targeting the protein of interest with either small molecules or antibodies¹¹⁵⁻¹¹⁷. Despite the efforts and the sophisticated specificity offered by anti-aggregation antibodies, these have so far failed at clinical trials because of the aberrant immune response they initiate^{118,119}. In this sense, small molecules offer substantial advantages over antibodies, since they are immunologically tolerated, more stable in the cellular environment, and often offer higher potential of crossing the blood-brain barrier¹²⁰.

In this work we presented anti-aggregating tryptophan-galactosylamine hybrid small molecules. We show a dual effect of these small molecules, capable of both reducing and reversing the formation of amyloid fibrils. Moreover, these molecules are non-toxic toward mammalian neuroblastoma and kidney cell lines and are able to significantly reduce the cytotoxicity induced by pre-formed fibrils of A β 42 or hIAPP. Collectively, the results here reported indicate that the hybrid system could be employed as a scaffold in the development of therapeutics towards multiple proteinopathies. Further studies will follow to obtain deeper insight on the mechanism of action of these hybrid molecules and to chemically improve them for higher efficacy and *in vivo* validation. Amyloids of A β 42 and hIAPP co-exist in major age-related diseases. Thus, the hybrid molecules could potentially offer single disease-treating therapeutics to these disorders.”

REVIEWERS' COMMENTS:

Reviewer #2 (Remarks to the Author):

The authors addressed my concerns, recommending for publishing.

Reviewer #3 (Remarks to the Author):

Brief summary of the manuscript:

In the paper titled "Tryptophan-galactosylamine conjugates inhibit and disaggregate amyloid fibrils of A β 42 and hIAPP peptides while reducing their cytotoxicity" the authors describe the use of the aromatic amino acid, tryptophan, to disrupt amyloid formation of the peptides Abeta42 and hIAPP. They improved upon the solubility of tryptophan through the creation of new derivatives with galactosylamine hybrids. The authors postulate that these hybrids, WGal, WGalNH₂, and WGalNAc, where WW serves as a negative control, are effective inhibitors of Abeta42 and hIAPP aggregation through the analysis of aggregation kinetics (ThT), circular dichroism (CD), electron microscopy (EM), and congo red birefringence (CR). Similarly, the authors use these hybrids in experiments of disaggregation of Abeta42 and hIAPP. Finally, the authors test the toxicity of hybrids then, in the presence of either amyloid, show the effect of hybrids on amyloid-induced cytotoxicity.

Overall impression of the revised manuscript

The authors have responded thoughtfully and comprehensively to my review. I particularly like the additional work and figures provided, especially in the cytotoxicity experiments.

Specific comments, with recommendations for addressing each comment

None.